# Driving the blue fleet: Temporal variability and drivers behind bluebottle (*Physalia physalis*) beachings off Sydney, Australia

Natacha Bourg[1¤a]*, Amandine Schaeffer[1,2], Paulina Cetina-Heredia[1¤b], Jasmin C. Lawes[3,4], Daniel Lee[1]

1 Coastal and Regional Oceanography Laboratory, School of Mathematics and Statistics, University of New South Wales, Sydney, New South Wales, Australia, 2 Centre for Marine Science and Innovation, University of New South Wales, Sydney, New South Wales, Australia, 3 Surf Life Saving Australia, Sydney, New South Wales, Australia, 4 School of Biological Earth and Environmental Sciences, University of New South Wales, Sydney, New South Wales, Australia

¤a Current address: Mediterranean Institute of Oceanography, Université de Toulon, Aix-Marseille Univ, CNRS, IRD, Toulon, France
¤b Current address: College of Natural and Computational Sciences - Department of Natural Science, Hawai'i Pacific University, Honolulu, Hawaii, United States of America
* natacha.bourg@univ-tln.fr

**Data Availability Statement:** Physalia physalis reports are in Supporting information. Wind measurement dataset can be obtained from the Bureau of Meteorology (http://www.bom.gov.au/

## Abstract

*Physalia physalis*, the bluebottle in Australia, are colonial siphonophores that live at the surface of the ocean, mainly in tropical and subtropical waters. *P. physalis* are sometimes present in large swarms, and with tentacles capable of intense stings, they can negatively impact public health and commercial fisheries. *P. physalis*, which does not swim, is advected by ocean currents and winds acting on its gas-filled sail. While previous studies have attempted to model the drift of *P. physalis*, little is known about its sources, distribution, and the timing of its arrival to shore. In this study, we present a dataset with four years of daily *P. physalis* beachings and stings reports at three locations off Sydney's coast in Australia. We investigate the spatial and temporal variability of *P. physalis* presence (beachings and stings) in relation to different environmental parameters. This dataset shows a clear seasonal pattern where more *P. physalis* beachings occur in the Austral summer and less in winter. Cold ocean temperatures do not hinder the presence of *P. physalis* and the temperature seasonal cycle and that observed in *P. physalis* presence/absence time-series are out of phase by 3-4 months. We identify wind direction as the major driver of the temporal variability of *P. physalis* arrival to the shore, both at daily and seasonal time-scales. The differences observed between sites of the occurrence of beaching events is consistent with the geomorphology of the coastline which influences the frequency and direction of favorable wind conditions. We also show that rip currents, a physical mechanism occurring at the scale of the beach, can be a predictor of beaching events. This study is a first step towards understanding the dynamics of *P. physalis* transport and ultimately being able to predict its arrival to the coast and mitigating the number of people who experience painful stings and require medical help.

places/nsw/MV07/observations/kurnell/). IMOS data is freely accessible at https://portal.aodn.org.au/.

**Funding:** The author(s) received no specific funding for this work.

**Competing interests:** The authors have declared that no competing interests exist.

## Introduction

*Physalia physalis* is commonly known by beach-goers as the Portuguese Man-Of-War in the Atlantic Ocean, and as the bluebottle jellyfish in the Indian and Pacific Ocean (also called the Indo-Pacific Portuguese Man-of-War). It is a colonial Cnidarian, part of the Order Siphonophorae, comprising interdependent, highly modified zooids that rely on each other to survive [1]. *P. physalis* is globally distributed, though is predominantly found in tropical and subtropical waters [2]. These pelagic organisms are effective predators that use their stinging cells to paralyze and feed on fish and fish larvae [2, 3]. *P. physalis* can be present in large swarms and since their tentacles deliver intense stings, this can have impacts on public health and coastal, commercial, and fisheries activities [4]. *P. physalis* stings are rarely lethal (only a few deaths recorded, e.g. [5]), but can cause lifelong scarring and systemic symptoms such as gastrointestinal, muscular, cardiac, neurological and allergic reactions [6, 7]. In Australia, many marine stings are treated by surf lifesaving personnel (surf lifesavers and lifeguards) who, between 2009/10 and 2019/20, have treated on average 40,128 stings each year, placing pressure on surf lifesaving service delivery and resources [8].

Community and economic impacts of *P. physalis* presence along the shore and *P. physalis* morphology are relatively well understood [2, 7], yet our understanding of drivers that affect *P. physalis* distribution, abundance, transport to the coast and their temporal variability is limited. Off the eastern coast of Australia, strandings of *P. physalis* typically occur more frequently in summer. This is consistent with the suggestion that colonies mostly reproduce in autumn, and have a lifecycle of approximately 12 months [9]. One of the dominant zooids in each *P. physalis* is the pneumatophore, which is filled with gas and enables the juveniles and mature specimens to float on the surface of the ocean, hence subject to ocean currents and waves. The pneumatophore is bilaterally flattened and acts as a sail, directly subject to the wind forces. The float exhibits a dimorphism, roughly half of the population have it tilted to the right, and the other to the left [1]. This causes an individual to drift at a certain angle, respectively to the left or to the right, of the wind direction [10]. This sail can be moderately contracted and erected by an individual *P. physalis* and is, along with the elongation and contraction of its tentacles, the only active influence a *P. physalis* has on its transport [2]. *P. physalis* cannot actively swim; thus, their distribution is uniquely driven by physical and environmental atmospheric and oceanographic conditions in conjunction with their specific biological traits.

Previous studies have linked *P. physalis* beach stranding to environmental conditions, and modelled their arrival to the coast, usually focusing on unusual events (e.g. swarms). [6, 11] studied massive beaching events that occurred in summer 2010 off the Basque coast (Spain) and the Mediterranean Basin using Lagrangian particle tracking. These beached *P. physalis* were determined to have originated from the northern part of the North Atlantic Subtropical Gyre, thousands of kilometers away from their beaching location [11]. However, [11] emphasized that the determination of the source is strongly dependent on the wind parametrisation used in the Lagrangian tracking model, and proposed wind as the dominant driver of *P. physalis* transport. [6] suggested that the massive arrival of *P. physalis* to the coast had been strongly influenced by an anomaly in zonal winds. Since then, massive beachings of *P. physalis* off the coast of Ireland in autumn 2016 (August and October) prompted further research [12] to identify source populations. Results suggested that the population of *P. physalis* may have originated from the North Atlantic Current, supporting the findings of [11]. Regarding the drivers of *P. physalis* transport in the Pacific Ocean, [13] have used sting reports from five summers across eight locations in New Zealand to develop a neural network-based model to simulate the arrival of *P. physalis* towards the shore, while assessing the contribution of large-scale winds and waves. Wave direction appears to influence transport far from the shore, while

wind direction and speed influence strandings close to the shore. [14] extended this analysis across New-Zealand and validated the influence of both wind and waves. They also highlighted that the meteo-oceanographic regimes responsible for beachings were location-specific. Unlike previous studies, a recent survey from [15] recorded nearly continuous strandings of *P. physalis* in Chile for three years, with the highest densities in the winter seasons two years in a row. These massive events coincided with an El Niño Southern Oscillation (ENSO) perturbation, with warmer ocean temperature conditions, and positive zonal wind anomalies (westerlies) transporting *P. physalis* to the coast, further highlighting wind as the major driver of *P. physalis* arrival to the coast.

The overarching goal of this paper is to extend our understanding of the environmental drivers of *P. physalis* by analysing strandings off three popular Australian beaches; our findings can assist coastal safety services towards the development of strategies that mitigate sting risks. We present the temporal variability of *P. physalis* beaching and sting reports over three locations off Sydney. We investigate the link between the presence of *P. physalis* on the shore and local winds, ocean temperatures, waves, ocean currents, and rip currents. Finally, the chances of *P. physalis* strandings are related to typical wind sectors, and discussed in lights of the coastline orientation.

## Data and methods

### Study area

Our study area is located on the southeast coast of Australia, and encompasses three beaches off Sydney that extend over ≈ 5.5 km of coastline, Clovelly (151.25˚E, 33.91˚S), Coogee (151.25˚E, 33.92˚S), and Maroubra (151.25˚E, 33.95˚S) (Fig 1). Maroubra is the most exposed and the longest beach (980m), and it is oriented directly towards the East. Coogee is smaller (410m) and more southward oriented. Note that Coogee has a small rocky outcrop (known as the Wedding Cake Island) 740m from the beach, which limits wave action on the beach. Clovelly beach is more South-oriented and is at the end of a narrow bay, hence more protected than the two other beaches (Fig 1).

### *P. physalis* datasets

We present two datasets that record the presence of *P. physalis*, beachings and stings. Beachings are recorded daily by the council lifeguards, written around 9AM for each beach, and are qualitative descriptions of *P. physalis* presence on the beach: "None", "Likely", "Some" or "Many". The dataset runs 4 years, from May 2016 to May 2020 (Fig 2). For this paper, we considered "Likely" days to be non-beaching days, and combined "Some" and "Many" to be observed. The resulting beaching dataset is then binary: 0 for absence, 1 for presence of *P. physalis*. At Coogee and Maroubra (Fig 2b and 2c), daily observational reports cover May 2016 and May 2020 with 94% and 93% data coverage (e.g. observations are missing on some public holidays when different life guards are on duty). In contrast, beaching reports at Clovelly beach only cover the warmer season, from October to April since the beach is not patrolled every day in winter (Fig 2a). Therefore, we focus the analysis of the beaching reports at Maroubra and Coogee, and look at composite conditions at Clovelly to understand differences amongst close-by locations.

In addition, we explore the variability of surf lifesaver sting reports for the same three sites. These reports list the number of people stung by *P. physalis* and treated by the surf lifesavers between 2016 and 2020, during the weekends and public holidays of patrol season (September—April). Estimates on beach attendance were also recorded from 2018 to 2020. For example, on a weekend day in summer, Coogee records more than four thousand people on the beach

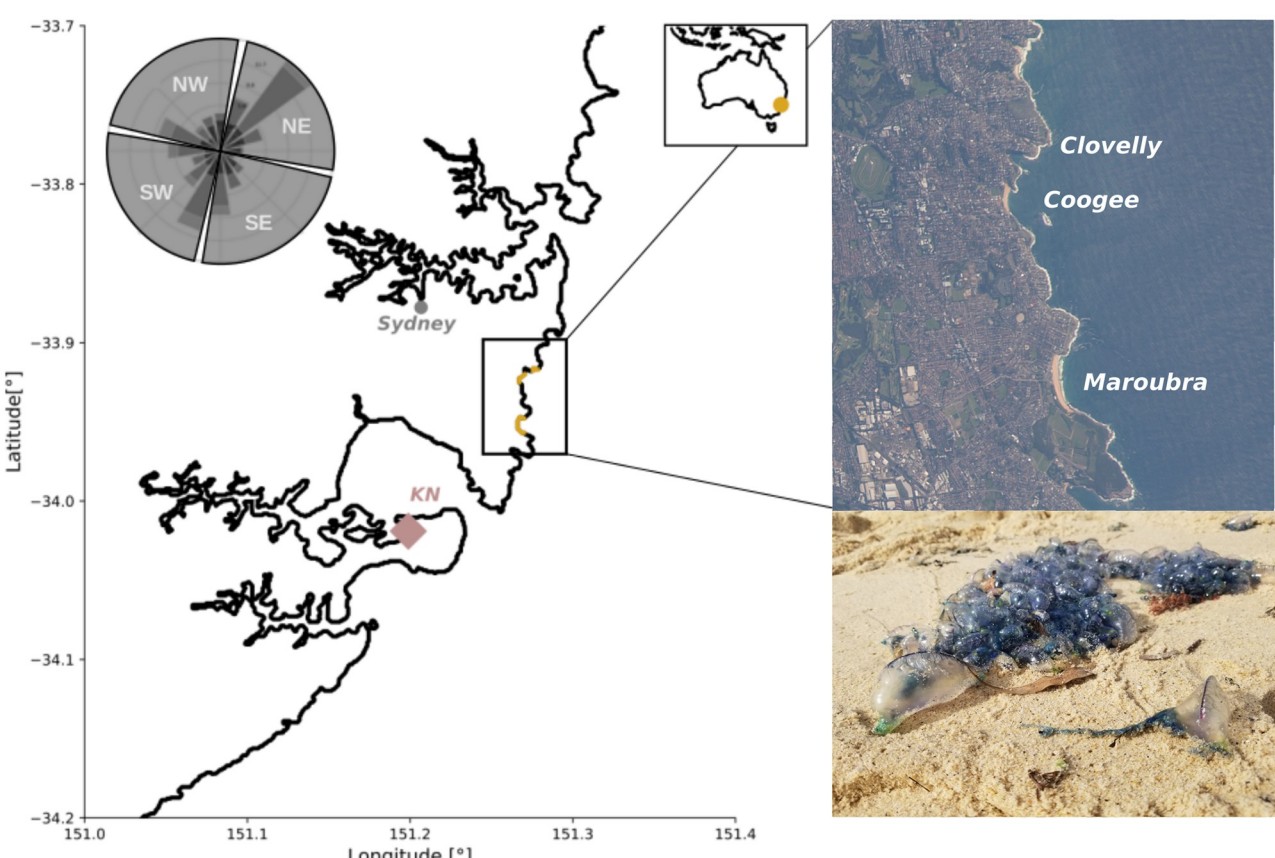

**Fig 1. Map of the study area off eastern Australia showing the location of the 3 different beaches (Clovelly, Coogee and Maroubra).** The location of Kurnell meteorological station is also shown (KN). The windrose in the top left shows the daily wind distribution measured at KN from 2016 to 2020 and the four wind sectors used in this study, which are roughly aligned with the local coastline. Top right: Satellite image of the different beaches (Image courtesy of the Earth Science and Remote Sensing Unit, NASA Johnson Space Center, eol.jsc.nasa.gov, Picture ID:ISS037-E-20021). Bottom right: picture of beached *P. physalis*.

and a maximum of 350 stings (02/01/2016). It should be noted that days when no stings were recorded does not equate to no *P. physalis* in the water. To remove false negatives, we do not analyse the data on days with no beach attendance. For matching days and locations (although different authors), beaching and sting datasets do not daily compare and sting reports are more frequent than beaching reports. Only 8%, 16%, and 32% of the stings corresponded to a beaching day at Clovelly, Coogee, and Maroubra, respectively. Due to this mismatch, and to the lack of data from April to September, the beaching dataset is the main material of the study and we use stings data to complete and nuance the analysis.

## Environmental datasets

To determine the influence of environmental parameters on the transport of *P. physalis* to the shore, we investigate the link between beaching temporal and spatial variability and those of other environmental variables which are known for their seasonality and/or influence on *P. physalis* transport: winds, ocean currents, water temperature, wave height and rip currents. Water temperature, rip currents and wave height data are estimated daily by the lifeguards. Wave height is described by six categories ranging from flat to very high wave height (< 0.5 meter, 0.5 meter, 1 meter, 1.5 meters, 2 meters, > 2 meters). Surface water temperature data

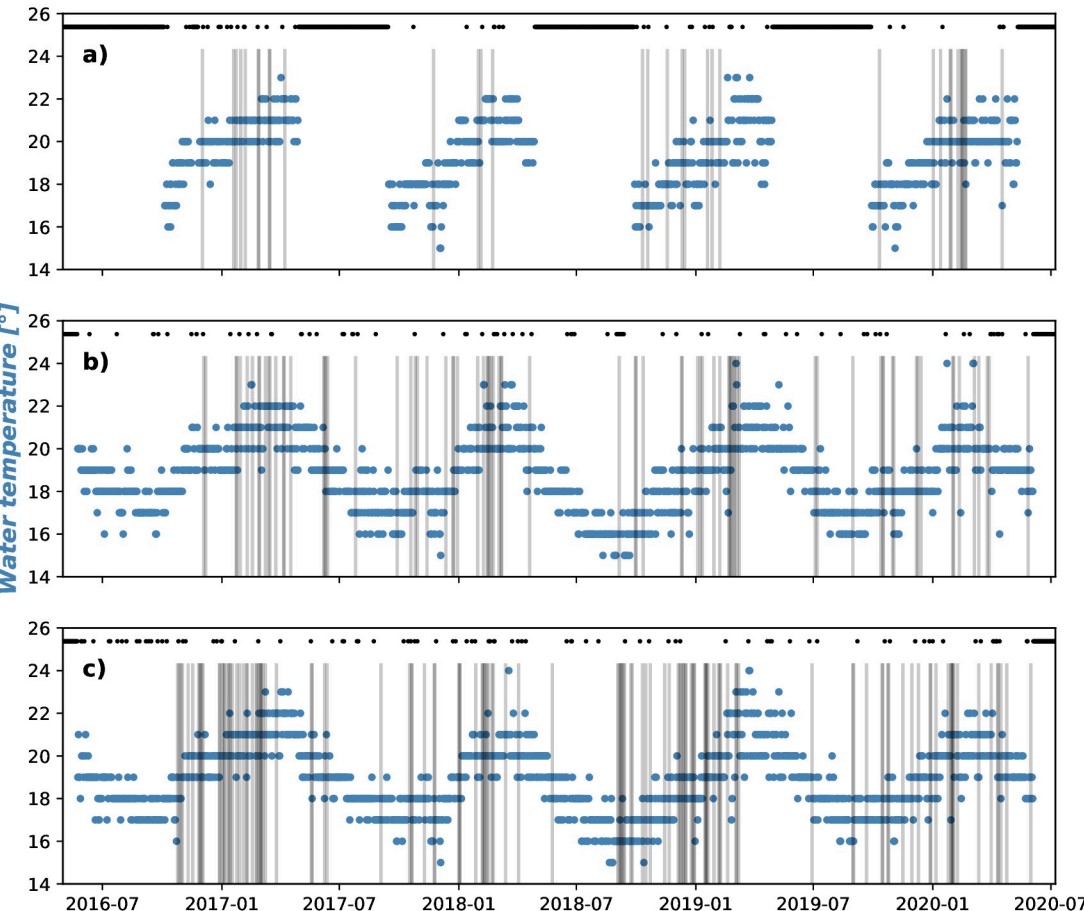

**Fig 2. Time series of the water temperature reported from 2016 to 2020 (Blue circles) for a) Clovelly, b) Coogee, c) Maroubra.** Days when beachings have been reported are shown by grey bars. Days when no beaching report is available (NaN) are scattered in black and shown at the top.

are recorded to 1˚C resolution (Fig 2). Rip currents estimates are qualitatively described by the lifeguards in three different categories. These ranks are then replaced by the arbitrary values of 0, 1, 2 respectively for "minimal", "be cautious", or "dangerous". Wind measurements were taken from the Kurnell weather station (ID: 66043) and used as a proxy for offshore winds (as per [16]). This station is located 8, 11 and 12 kilometers away from Maroubra, Coogee and Clovelly beaches respectively (Fig 1). Wind data are recorded every half an hour. The wind zonal and meridional components are daily averaged starting at 9AM local time for the beaching reports, and from 5PM local time for the sting reports, to match the timing of the observations. Predominant winds in this area are north-easterly, westerly and southerly, as shown on the windrose in Fig 1. For further detail on the monthly variability of winds, we refer to Fig 7 and [17].

Local ocean currents time-series are also considered. We analyse ocean current velocity data from close-by moorings along the coast. One mooring is located above the 100 m isobath 2 km from the shore, and another above the 65 m isobath 10 km from the shore (SYD100 and ORS065, respectively, described in [18, 19]). The mooring's instruments measure U (zonal) and V (meridional) current velocity components throughout the water column every 5

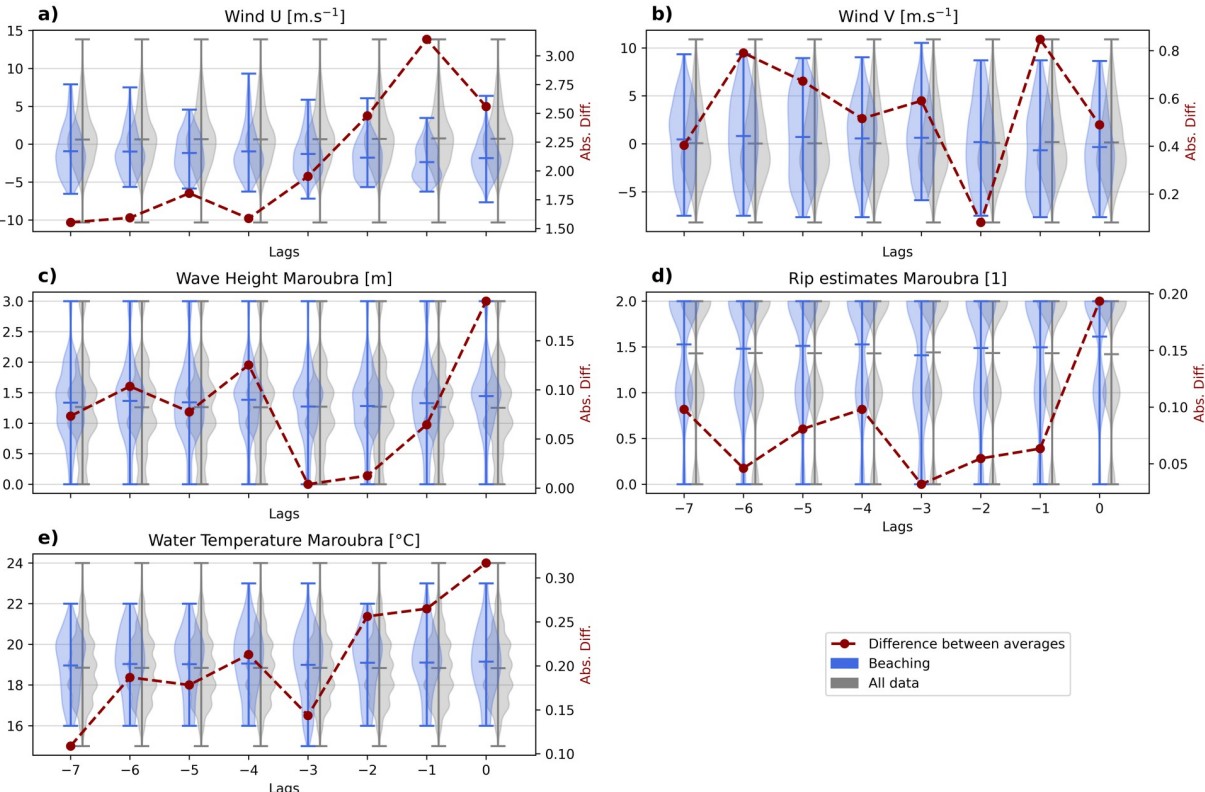

**Fig 3. Each subplot shows two violinplots of a variable (U wind, V wind, wave height, rip currents, water temperature) with all data (grey), and data at the condition of a beaching at Maroubra λ days before (blue), for λ going from -7 to 0.** For each lag λ, the difference between the average of the two violinplots (beaching condition—all data) is plotted in red dashed line.

minutes and every 4–8 meters in depth. Here, we used daily averages at the shallowest bins (11 m and 12 m, respectively).

We identify the temporal lag for which each variable is influencing the beaching of *P. physalis*, between λ = -7 to 0 days before the latter observations. Fig 3 shows the difference between the distribution of each variable when considering all data, or a subset when a beaching was recorded λ days later. We consider that the greater is the difference, the stronger is the relationship. The wind influence appears to be maximum for a lag of one day (Fig 3a and 3b: the maximum of the red line is at λ = -1), while considering other variables the same day as the beaching seems appropriate (Fig 3c–3e: the maximum of the red line is at λ = 0).

## Statistical analysis

Taking into account the strong auto-correlation of time-series, we use a Generalised Estimating Equations (GEE) [20] model with an autoregressive AR(1) structure. We use this method, not to create a model able to predict the arrival of *P. physalis* to the shore, but to identify statistically significant relationships between environmental variables and the coastal presence of *P. physalis*. The algorithm is run in a backward step-wise fashion so as to only keep relevant variables. The response is the binary beaching event variable at Maroubra concatenated to the equivalent variable measured at Coogee. The predictors are the lagged wind zonal (cross-shore) and meridional (along-shore) components, the water temperature, wave height, rip currents estimates and the ocean currents zonal and meridional components, as well as a variable

accounting for an annual cycle peaking on the the 7th of February, when the maximum beachings was observed, to represent seasonality. It is defined as: $seasonality = 1 + cos\left(2\pi \frac{dayofyear - maxbbday}{365}\right)$, where *dayofyear* is the day of the year (i.e. 31/12: 365, 01/01: 1. . . etc) and *maxbbday* is the day with the highest number of beachings on average over the four years (7th of February). A "site" variable (1 for Coogee, 2 for Maroubra) is also included in order to identify differences between the two locations (length, shape, orientation). We report Wald tests using a naïve variance estimator.

The same analysis is run with stings data as the response variable. The time-series of the number of stings are turned into binary data (0: no stings, and 1: at least 1 stings). We use the same predictor variables than for the beachings, and add the root square of the beach attendance time-series (the root square is taken to respect the linearity of the logit assumption).

## Results

### Temporal variability of *P. physalis* beaching

Between 2016 and 2020, daily observations of beachings from the lifeguards show that the occurrence of *P. physalis* off Sydney varies both temporally and spatially (Figs 2 and 4), with the beaching frequency differing from one beach to another. Maroubra is where *P. physalis* is the most likely to be sighted, with 132 beaching events over four years, followed by Coogee with a total of 82, and Clovelly with 38 beaching days (October to April only). These differences can be related to differences in the beach lengths and exposures to open ocean. Maroubra is the largest, followed by Coogee, while Clovelly is far narrower than the others. In addition, the Wedding Cake Island located in front of Coogee, and the enclosed geography of Clovelly, may prevent the arrival of *P. physalis* (Fig 1). Simultaneous beachings in Maroubra and Coogee occur only 14% of the beaching days. However, for the sting report dataset, this number increases to 54% of simultaneous stings at the two beaches, and the correlation between the two time-series of the number of stings is significant $r = 0.3$ ($p < 0.0001$).

For Coogee and Maroubra, where daily data are available all year long, beaching events display a strong seasonal signal, with frequent events in the Austral summer (December January

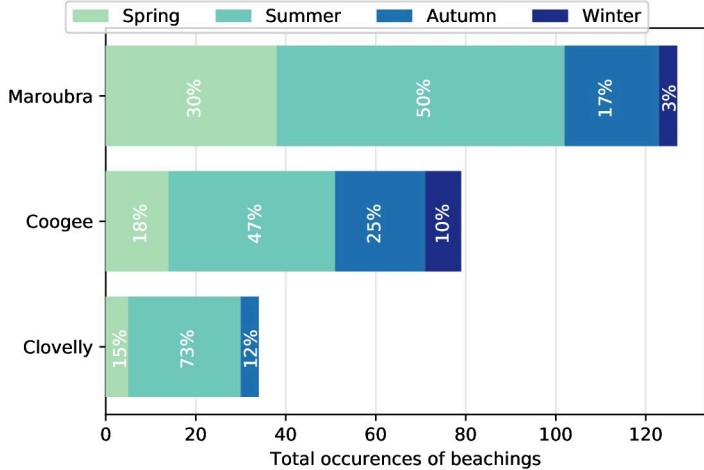

**Fig 4. Bar plot for each beach, showing the occurrence of observed beaching events for each season (see legend) from the 2016–2020 daily lifeguard data.** The total occurrences indicate the number of days over the four years and the percentages for each season are relative to the total numbers of occurrences per beach (e.g. in Maroubra 50% of the beachings occurred in summer). Winter months are not seen for Clovelly as there is no data.

February) and very infrequent events in the Austral winter (June July August) (Fig 4). Indeed, between 2016 and 2020, 50% and 47% of strandings occurred during the three months of summer in Maroubra and Coogee respectively. In Maroubra, spring is (after summer) the second season with most beaching events (30% of beachings), whereas in Coogee, beaching events are more numerous in autumn (25%) than spring. Interestingly, there are still instances of winter beaching for Coogee and Maroubra, up to 10% of annual sightings in Coogee, and 3% in Maroubra. This suggests that despite the seasonal cycle to their strandings, *P. physalis* survive winter time and cold temperatures and can still be advected to the coast. Therefore, the seasonality of their strandings is likely influenced by environmental parameters rather than only driven by their lifecycle.

## Drivers of *P. physalis* transport to shore

Since *P. physalis* is more common in summer, but not impossible in winter (hence still alive), the question is whether environmental variables drive its seasonality. Fig 5 provides a view of seasonal cycles of beaching events at Maroubra (Coogee in Supp. Mat., S1 Fig) for each week of the year (averaged over the four years), together with water temperature, cross-shore winds and wind speed. Although the water temperature displays a strong seasonal signal (Fig 5a), it does not have the same phase as the seasonal signal of beachings (shown by the grey bars) which peak in early February, while ocean temperatures are maximum in late March. However, the seasonality of wind direction visually matches the annual cycle in *P. physalis* beaching. In particular, while the wind speed shows no seasonal cycle comparable to the beaching variability (Fig 5c), the weekly mean of the cross-shore component of the wind shows negative values, hence a wind blowing towards the shore (easterlies) in the first and last 12 weeks of the year, when sightings of *P. physalis* are frequent. Conversely, positive values, hence a wind blowing predominantly from land (westerlies) is dominant in winter when *P. physalis* rarely reach the coast (Fig 5b). The maximum sightings also occur during the strongest easterly wind (weeks 6 and 52) and no beaching occurred during the strongest westerlies (weeks 28–30, 32–33).

The link between environmental variables and the spatial and temporal variations in *P. physalis* arrival to the shore is further investigated using a GEE model. Tables 1 and 2 show the variables having a statistically significant relationship to beaching and sting events, respectively.

This analysis reveals a significant relationship between *P. physalis* beaching and winds, rip currents and an annual cycle. In particular, cross-shore wind was identified as the main driver for beaching events, with a high coefficient and the lowest p-value in the model outputs (Table 1). The negative coefficient for cross-shore wind shows that negative zonal winds (i.e. towards the shore) are likely to lead to a beaching event according to the model. These statistical results support the visual match shown in Fig 5. In increasing p-value order: cross-shore wind, rip currents, the annual cycle variable and along-shore winds all contribute to the temporal variability of *P. physalis* beachings. It should be noted that information on the site location (Coogee or Maroubra) does not seem to improve the model. Water temperature is not an important variable of the model either, which is consistent with beaching events that have occurred at both the coldest (16˚C) and warmest (23˚C) water temperatures (Fig 2), hence water temperature within this range (16–23˚C) does not prevent nor significantly drive beachings. Although ocean currents are thought to play a role in *P. physalis* transport and regions of origin [11, 12], we find no clear pattern between ocean current velocity and observed beaching events. The same holds for wave height.

Regarding sting events, the beach attendance, the site, and the zonal wind time-series are the main variables to model the variablity of the stings in summer (Table 2). Unlike for

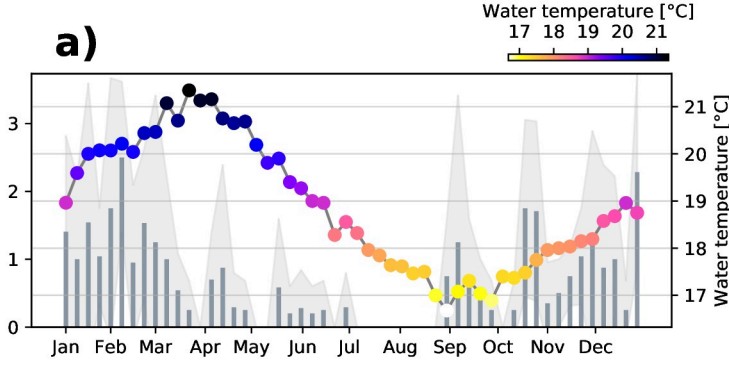

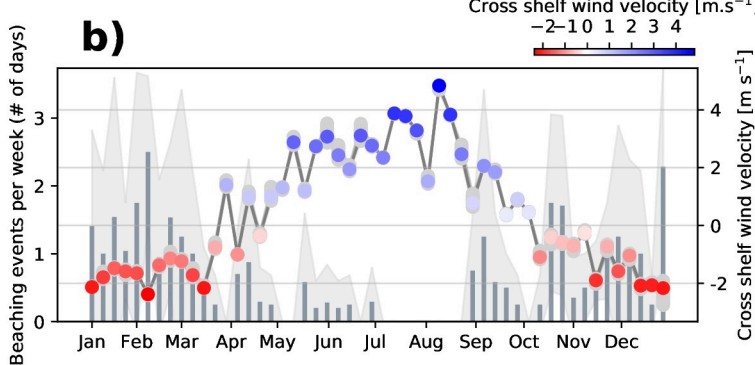

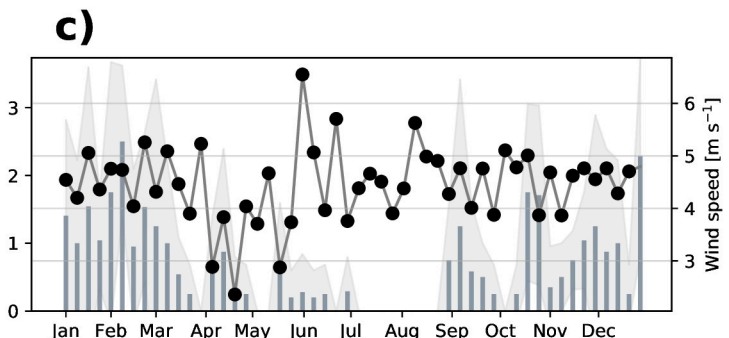

**Fig 5. Weekly climatology of beaching events and environmental variables at Maroubra.** Grey bars on all panels show the number of beaching events per week over 2016–2020 and the standard deviation is shown in light grey shading. In panel a, the weekly mean water temperature is overlaid (right axis and colours). In panel b, the weekly mean cross-shore wind velocity component is overlaid (right axis and colours) with positive (negative) values showing wind from (towards) the coast. In panel c, the mean weekly wind speed is overlaid (right axis).

**Table 1. Outputs of a backward step-wise GEE analysis with the binary beaching event variables at Maroubra and Coogee concatenated as the response variable.** Predictors are the lagged wind zonal and meridional components, the water temperature, wave height, rip currents estimates, the ocean currents zonal and meridional components, a *seasonality* variable accounting for an annual cycle peaking on the 7th of February, and a "site" variable (1 for Coogee, 2 for Maroubra). Only significant predictors are shown with their coefficients, 95% confidence interval, and p-value.

| Predictors | Coefficient | 95% CI | p-value |
|---|---|---|---|
| Wind U velocity | -0.32 | (-0.40;-0.24) | 0.000 |
| Rip currents | 0.32 | (0.12;0.52) | 0.002 |
| Seasonality | 0.41 | (0.10;0.73) | 0.010 |
| Wind V velocity | 0.06 | (0.01;0.11) | 0.021 |

**Table 2. Outputs of a backward step-wise GEE analysis with the binary sting event at Maroubra and Coogee concatenated as the response variable, in summer only.** Predictors are the lagged wind zonal and meridional components, the water temperature, wave height, rip currents estimates, the ocean currents zonal and meridional components, a *seasonality* variable accounting for an annual cycle peaking on the 7th of February, a "site" variable (1 for Coogee, 2 for Maroubra) and the root square of the beach attendance time-series. Only significant predictors are shown with their coefficients, 95% confidence interval, and p-value.

| Predictors | Coefficient | 95% CI | p-value |
|---|---|---|---|
| Wind U velocity | -0.28 | (-0.42;-0.14) | 0.000 |
| Beach attendance | 0.01 | (0.005;0.02) | 0.001 |
| Site | -0.95 | (-1.66;-0.24) | 0.009 |

beaching events, only the zonal component of the wind has a p-value < 0.05 and rip currents do not seem to be driving sting events. Rip currents may then be an important process that transports the *P. physalis* from nearshore waters to the beach itself, but does not affect stings, which might occur in the ocean, to the same extent.

Since wind appears to be the main driver for *P. physalis* arrival to the shore, we investigate the composite wind conditions for beachings and stings. Focusing on summer, when the majority of sightings occur, rose plots of wind conditions for 24 hours preceding *P. physalis* sightings show that north-easterly and south-easterly (i.e. shoreward winds) are the two most favourable directions for *P. physalis* beachings and stings (Fig 6), while no beaching occurs from south-westerly winds. There are spatial differences between the locations: North-East is the most favourable wind condition for beaching at Coogee and Maroubra, while it is South (-East) for Clovelly. Sting and beaching reports show similar favourable conditions for *P. physalis* in Maroubra and Coogee, but a few differences in Clovelly (Fig 6). At Clovelly, beachings are reported during north-easterly and southerly winds (consistent with the beach orientation), but fewer stings are reported during the latter.

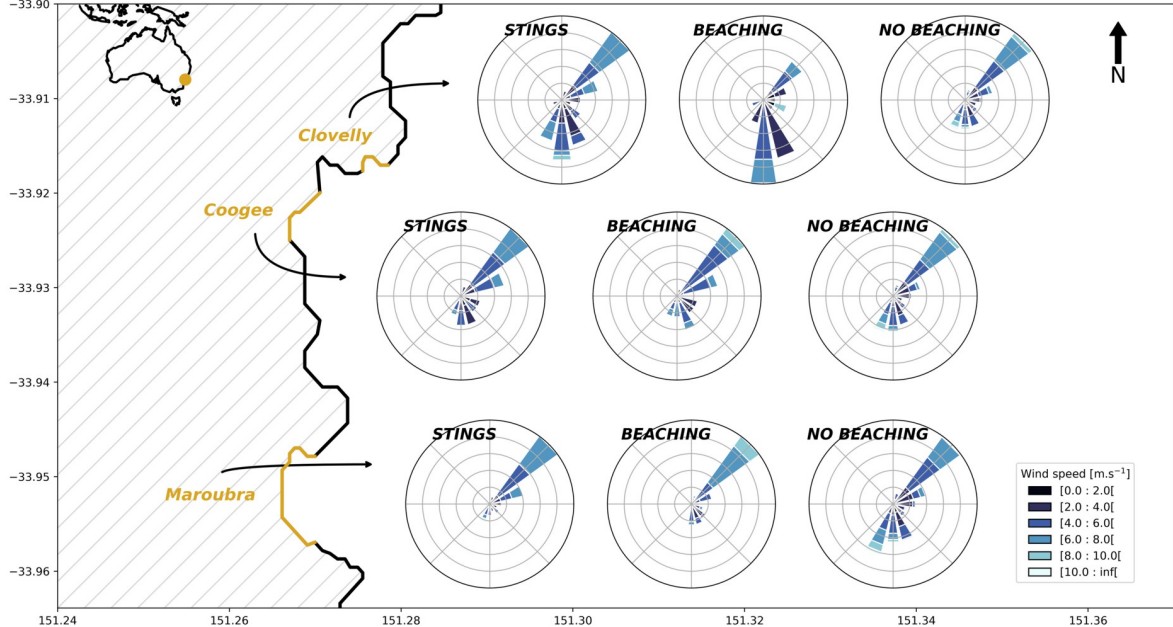

**Fig 6. For each beach: Summer rose plots showing daily wind conditions a day prior to sting (Stings) reports, beaching (Beaching) and no beaching observations (No Beaching).** Each beach is shown with the yellow line.

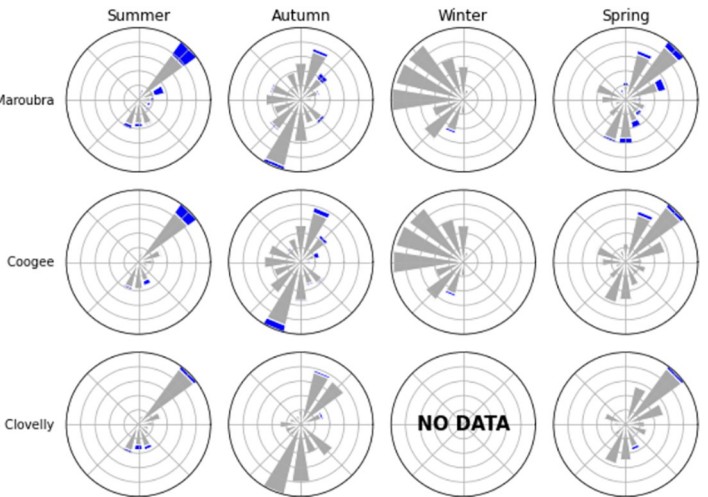

**Fig 7. For each season and each site, a rose plot of wind conditions is shown.** The blue part represents the portion of dates with these wind conditions and a beaching event.

## Chances of *P. physalis* beachings and stings per beach and wind sector

We now investigate how likely it is to see *P. physalis* during favourable wind conditions and how the likelihood varies between seasons and sites. For each beach, Fig 7 displays a windrose of the wind conditions of each season, with the blue color showing the proportion of *P. physalis* beaching events. Wind conditions display a strong seasonality, with summer dominated by (north-)easterly winds (favourable for beachings, see Fig 6), and winter by westerly winds (unfavourable for beachings, see Fig 6). Spring and Autumn can be seen as transitional in term of wind conditions. It is important to note that on top of the seasonal variability of winds, the proportion of beaching events for each wind sector differs from a season to another. We quantify the percentage of instances when each wind direction brought *P. physalis* to the shore in Table 3. Overall, for any wind condition, Maroubra is where beachings are most likely, followed by Coogee and Clovelly (Table 3). Again, these differences are consistent with the geomorphology of each beach shown on Fig 1. At Maroubra and Coogee, most of the beaching events are associated with north-easterly and south-easterly winds the day prior to sightings (Fig 7), with similar chances of beaching (i.e. 16–17% for Maroubra, 10–12% for Coogee, Table 3). Focusing on summer only, the north-easterly seabreeze is slightly more likely to lead to a beaching event in Maroubra, with a 24% chance of *P. physalis* beachings, while it is south-easterly wind that is more prone to beaching at Coogee (13%). For Clovelly as well, south-easterly winds (12%) are more favourable than north-easterly winds (4%). For all three locations,

**Table 3. Frequency of beaching events per wind sector.** The 4 sectors: North-East (NE), South-East (SE), South-West (SW) and North-West (NW) are redefined following the orientation of the coastline (see windrose of Fig 1). Blue: computed on data 24 hours before a sighting, all year round from 2016–2020, black: on summer dates only from 2016–2020. The number of days for each wind sector is indicated in bracket (all year; summer only).

| Chances of beaching when | Clovelly | | Coogee | | Maroubra | |
|---|---|---|---|---|---|---|
| NE (368; 167) | X | 4% | 10% | 11% | 17% | 24% |
| SE (260; 111) | X | 12% | 12% | 13% | 16% | 22% |
| SW (297; 53) | X | 6% | 2% | 5% | 2% | 0% |
| NW (211; 10) | X | 0% | 1% | 0% | 1% | 0% |

**Table 4. Frequency of summer stings per wind sector.** Computed on data 24 hours before a report, on summer months only, from 2016 to 2020. The 4 sectors: North-East (NE), South-East (SE), South-West (SW) and North-West (NW) are redefined following the orientation of the coastline (see windrose of Fig 1). After each wind sector the number of instances that wind blew from this direction is written in parentheses. NW results will not be considered as there are only 2 reports available for this wind direction.

| Chances of stings when | Clovelly | Coogee | Maroubra |
|---|---|---|---|
| NE (73) | 56% | 43% | 55% |
| SE (48) | 48% | 42% | 38% |
| SW (21) | 24% | 5% | 10% |
| NW (2) | 50% | 0% | 50% |

chances of beachings during westerly winds (North-West and South-West) are negligible, showing that winds from the coast usually prevent the arrival of *P. physalis* to shore. We note that favourable wind conditions for *P. physalis* beaching in Maroubra and Coogee (i.e. North-East followed by South, Fig 6) are more frequent than those for beaching in Clovelly (i.e. South followed by North-East, Fig 6), further explaining the differences in *P. physalis* abundance between the locations shown in Fig 4.

Even if the number of stings is likely underestimated (not all stings are treated and hence recorded by the lifeguards), the chance of stings displayed in Table 4 shows more frequent stings than beachings for any wind condition. This could be due to the differing datasets, but also to the fact that beachings *P. physalis* are only reported by lifeguards at 9AM, while stings can occur over a larger area (on the beach and in adjacent water) during the whole day. Still, the wind conditions transporting *P. physalis* to the shore are qualitatively the same for the two datasets. Indeed, as with the beachings, it is unlikely for stings to be reported after westerly winds for all three locations. For Coogee and Maroubra, North-East is the most favourable condition followed by South-East, while it is the opposite for Clovelly. Interestingly, north-easterly and south-easterly winds have almost equal chances to be followed by beachings at Coogee and Maroubra, although the chance of stings is much higher during north-easterly than south-easterly wind conditions, especially for Maroubra. As already mentioned (Table 2), there is an association between sting reports and the presence of people recreating in the water which can be influenced by the weather conditions, with north-easterly winds often associated with warm sunny days, while south-easterly are often the result of a low pressure system with rain and swell.

## Discussion

This study is the first one to explore *P. physalis* beaching observations in relation to various environmental variables in Australia. Using multiple datasets collected from three proximate locations off Sydney's coast, our results show that the occurrence of beachings differ in time as well as from one beach to another. We also demonstrate a strong relationship between this spatio-temporal variability and wind direction, with North-East and South-East clearly identified as favourable wind directions for *P. physalis* beachings at these locations. The differences in occurrence of observed beaching events among close-by beaches are likely explained by the geomorphology of the coastline as well as by the differences in frequency of favourable wind conditions. For example, winds favourable for beachings at Clovelly are different from the two other beaches (Fig 6; Table 3), and it could be explained by the orientation of Clovelly, which is oriented more towards the South than the other sites. The year-round dataset over four years enables the identification of a clear seasonal pattern in the frequency of beaching events. Most *P. physalis* beachings occurred in summer, with the least number of beachings recorded in winter. Given results from the GEE analysis, Figs 5 and 7, this variability seems strongly forced by the wind's seasonality.

When analysing the beaching and sting datasets, some of the findings were unexpected. There are days where *P. physalis* beachings were sighted while no stings were reported and vice-versa. As the number of stings depends on the number of people present in the water, no stings being reported does not necessarily mean that no *P. physalis* were present in the water. Differences between the two datasets could be explained by the difference in the timing of the reports but also by the nature of the reports (stings happen in the water, while beachings are reported only when *P. physalis* are stranded on the shore). The discrepancy may also be due to weather conditions: north-easterly winds usually occur on sunny days, while southerly winds are often grey and rainy, influencing the number of days with beach-goers and their exposure to stings. Also, north-easterly winds usually occur in the afternoon at these locations when beach attendance is high and stings more likely, while beachings are recorded in the mornings by lifeguards. Even if sting and beaching reports do not match on a daily basis, the results regarding their link to environmental variables are quite similar across both datasets. We also identified North-East and South as typical wind conditions when stings occurred, and the differences between beaches and wind directions were similar to results using the beaching reports. Hence, wind direction is proposed as a major driver of seasonal patterns observed in *P. physalis*' arrival to shore although the impact of their life cycle cannot be ruled out.

The relationship between winds, surface ocean currents, and *P. physalis* movements was first investigated by [21]. They studied *P. physalis* drifting direction versus the wind and observed a clear tendency of *P. physalis* moving at around 45˚ of the wind direction. The drifting angle of *P. physalis* and its asymmetry was later extensively studied by [1] using field observations and conducting experiments. They found that left (right) handed *P. physalis* drifted at 40˚ to the right (left) of the wind direction under light winds (under $< 8$ m s$^{-1}$), and drifted in the direction of the wind under stronger winds. These concepts were extended by [10, 22], when a theory regarding *P. physalis* transport was developed by comparing its hydrodynamics to a wind-powered sailboat. This study shows the complexity of *P. physalis* hydrodynamic relationship with winds. *P. physalis* drifting angle to the wind is now believed to be approximately 40˚ and is suggested to vary with wind speed and with the size of *P. physalis* [1, 22]. Due to the lack of data on *P. physalis*'s handedness, this has not been investigated in the present study, but a suggested hypothesis is that of the two favourable wind directions identified, one direction (e.g. North-East) will push left-handed *P. physalis* to the coast; while the other (e.g. South) will push the right-handed to the coast.

It is important to highlight that the wind is a major contributor to the ageostrophic component of the surface current (influencing circulation and generating local waves). Stokes drift and wind-induced currents are known to be highly relevant in regard to the transport of passive tracers in the ocean surface [23, 24]. Thus, this relationship between beaching events and cross-shore wind can be explained by the wind drag on the above-water sail, but also by the wind-influenced water transport.

Some beaching events are recorded during winter months dominated by westerly winds, for example 10% of beaching events off Coogee occurred in winter (Fig 4). In addition, there is a high frequency of beaching events in spring recorded during weeks dominated by south-westerly winds, as can be observed during September and May in Fig 5. If wind was the only driving variable, beachings would not be expected when wind is coming from land. Moreover, we found that beachings are more likely in summer under any wind conditions (Table 3), and chances of beaching events under a certain wind direction vary from one season to another (Fig 7). On top of that, the variable accounting for the seasonal signal appears as a significant variable in the GEE model (Table 1), meaning that the seasonality of beaching events is not entirely captured by winds and rip currents. The wind is therefore not the sole driver of *P. physalis* transport to shore and other physiological or environmental variables such as sea state

and ocean currents could also influence *P. physalis* transport and subsequent beaching events. Stokes drift, the movement caused by wave propagation, can also have an important role on the drift of organisms and inert particles in the ocean (i.e. lobster larvae [25], plastic [26]), and is likely relevant to *P. physalis* transport. However, our attempt to link ocean current and wave height with *P. physalis* beachings was not conclusive. This could be due to the datasets, with qualitative wave height data that may not be precise enough to resolve the scales at stake. Moreover the shallowest measurement available of ocean currents is located at a 11m depth, while the main body of *P. physalis* colonies usually only reach few centimeters. Observations of ocean currents closer to the surface and of higher resolution may be necessary to expose any dependence of beaching events on these variables.

To date, little is known about the ecology, lifecycle, and pathways of *P. physalis*. It has been suggested that colonies have a lifecycle of approximately 12 months [9] but specific details are lacking. Environmental factors such as light, temperature, salinity and food availability may have an effect on jellyfish reproduction and growth rates [2, 9, 27]. The results presented in this study demonstrate that *P. physalis* can be present close to the coast year-round and abundances may fluctuate during the year but also from one year to another, but show a general seasonal cycle which could be due related to *P. physalis*'s lifecycle. We do not find local ocean temperature to be a predictor of beaching events, and, within the range reached at these latitudes, cold water is not preventing the presence of *P. physalis* (e.g. beachings observed on the 01/09/2019 with water at 16˚C). However, the seasonal cycle of ocean temperature at Sydney lagged by 3–4 months (similar to temperature at lower latitudes) is correlated with beaching events. We therefore do not exclude the possible influence of sea surface temperature on *P. physalis*'s abundance offshore [28, 29]. Our results show that local winds are important in nearshore waters, but we also hypothesise that ocean circulation offshore may be important in transporting *P. physalis*, for example from tropical zones to the temperate latitudes (where our observations are from), with the East Australian Current (EAC) being the main pathway [30]. Still, the unknown source location and variability in offshore abundance of *P. physalis* is a major source of uncertainty in this study and a clear knowledge gap to be addressed in future research.

It should be kept in mind that the observational dataset is anecdotal, and reliability of counts assessing *P. physalis* beaching may be affected by human subjectivity. Additional data collection would ideally include *P. physalis* size and morphology (left or right-handed). Records of an estimated number of *P. physalis* beached, as well as sustained observations at more locations would also be beneficial.

To conclude, our four year observational database of *P. physalis* beachings off Sydney, showed a clear seasonal signal of beachings in this area, with most beaching events occurring in summer. We identified a strong dependence of beaching events with wind direction at seasonal and daily timescales. These results are in agreement with literature suggesting that the wind plays an important role in *P. physalis* transport in other study areas (e.g. [6, 11, 13, 15, 28, 29]). Interestingly, rip currents, a physical mechanism occurring at the scale of the beach, are positively correlated to the beaching of *P. physalis* while stings are related to the sites. We expect these results to be valid for *P. physalis* arrival to the coast in other locations. However, the role of other variables need to be further investigated when more data are available, in particular for unexpected and extreme beaching events.

## Supporting information

**S1 Fig. Weekly climatology of beaching events and environmental variables at Coogee.** Grey bars on all panels show the number of beaching events per week over 2016-2020 and the standard deviation is shown in light grey shading. In panel a, the weekly mean water

temperature is overlaid (right axis and colours). In panel b, the weekly mean cross-shore wind velocity component is overlaid (right axis and colours) with positive (negative) values showing wind from (towards) the coast. In panel c, then mean weekly wind speed is overlaid (right axis).
(TIFF)

**S1 Data.**
(XLSX)

## Acknowledgments

The authors would like to thank Duncan Rennie, Manager Public Safety and Aquatic Services, as well as the Randwick City Council and all lifeguards and surf lifesavers who collected the datasets. The authors would also like to thank Stats Central from UNSW for their guidance with novel statistical analyses and the anonymous reviewers for their invaluable feedback during the preparation of this manuscript for publication.

## Author Contributions

**Conceptualization:** Amandine Schaeffer, Paulina Cetina-Heredia.

**Data curation:** Amandine Schaeffer, Jasmin C. Lawes, Daniel Lee.

**Formal analysis:** Natacha Bourg.

**Investigation:** Natacha Bourg.

**Methodology:** Natacha Bourg, Amandine Schaeffer, Paulina Cetina-Heredia.

**Project administration:** Amandine Schaeffer.

**Supervision:** Amandine Schaeffer, Paulina Cetina-Heredia.

**Writing – original draft:** Natacha Bourg.

**Writing – review & editing:** Amandine Schaeffer, Paulina Cetina-Heredia, Jasmin C. Lawes, Daniel Lee.

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
