## [Decision Letter · Decision Letter 0]

4 May 2021

PONE-D-21-06875

Driving the blue fleet: Temporal variability and drivers behind bluebottle Physalia physalis beachings off Sydney, Australia.

PLOS ONE

Dear Dr. Bourg,

Thank you for submitting your manuscript to PLOS ONE. After careful consideration, we feel that it has merit but does not fully meet PLOS ONE’s publication criteria as it currently stands. Therefore, we invite you to submit a revised version of the manuscript that addresses the points raised during the review process.

Before this ms can be published, substantial changes need to be implemented to augment the analyses and strengthen the main lessons of the paper.

The introduction needs to be streamlined. For instance, the second paragraph (lines 17 to 46) is very lengthy and could be split into smaller focused sections.   The next two paragraphs (lines 47 to 94) could also be streamlined, and some of the material could be moved to the discussion, where it would be placed in context of the project’s findings.

As stated by one reviewer, the authors have not performed the proper statistical analysis to sustain their conclusions. Even though several papers dealing with the analytical approach required to understand environmental forcing are cited, the authors have only used person correlations in their analysis. The authors have to improve this section (by including a whole Data Analysis section in the methods) and undertake a more comprehensive analysis of the data at hand.to determine the influence of these factors (and potentially their interactions) on the beachings (and the summer stings).  The current piece-meal approach, where a single variable is considered at a time need to be augmented and strengthened. 

To facilitate the understanding of the patterns, I would also suggest focusing on the two sites with year-long data and removing the third site (rocky shore with only summer-time data).   Limiting the analysis to the two sites with year-long data (Clovelly and Maroubra) provides a more comprehensive and comparable perspective.  The ms already explains that this site is inherently different: “Note that Coogee has a small rocky outcrop (known as the Wedding Cake Island) 740 m from the beach, which limits wave action on the beach. Clovelly beach is more South-oriented and is at the end of a narrow bay, hence more protected than the two other beaches (Fig 1).”

Additionally, the Physalia physalis datasets need to be analyzed in a more quantitative fashion.  In particular, I would suggest the following analyses:

* Number of beachings:

Compare the number of beaching observations versus the number of survey days from a beach to beach.  There are 38 and 132 beaching reports for Clovelly and Maroubra respectively, even though the two beaches were surveyed on 94% and 93% of the days. Is this difference significant?  Is there an overall higher beaching rate in Maroubra?  Despite the data gaps, I would suggest you perform a cross-correlation to quantify how well the beachings data at the two beaches cross-correlate with each other. 

* Number of stings: 

I would suggest focusing this analysis on the same two beaches used in the beachings analysis, and discarding the data from Coogee.  Despite the data gaps, I would suggest you perform a cross-correlation to quantify how well the beachings data at the two beaches cross-correlate with each other. 

You state that “More than 10 stings have been reported 6, 9 and 10% of all patrolled days for Clovelly, Coogee and Maroubra”. 

Why did you not consider days where less than 10 stings have been recorded?  You could use values above and below this threshold as two separate categories (low and high), or you could take the log10-transform of the data.

How was this threshold number selected?  Seems like anomalous events should be determined on a beach-basis, not using the same threshold across all beaches.   I would suggest you provide a data summary of the number of stings reported per day, and then attempt to model these distributions to figure out “outlier days” for each beach. 

* Number of beachings VS Number of stings:

It would be very useful to investigate whether these two datasets are correlated.  Using the summer-period only, when stings are reported, can you perform a correlation for each beach, to see if there are more stings on days with more beachings.  This would be a very informative analysis.

* Wind Data: 

Can you please define the wind sectors and provide some summaries of wind speed / direction for the different seasons?  The ms currently states “predominant winds in this area are north-easterly, westerly and southerly, as shown on the windrose in Fig 1.”

The analyses of beachings per wind direction also need to involve statistical tests, using either chi-square tests or logistic regressions.  Reporting mere proportions is not enough.  You need to provide a sense of the variability (SD for the proportions) and the associated p values.  

* Ocean Currents:

Can you please report how well the near-surface and the integrated currents correlate with each other?  And report how well they match the wind speeds?   Currently, the ms states: “Here, we used daily averages at the shallowest bins (11 m and 12 m, respectively) and the depth integrated flow”.

* Seasonality:

The proportion of beachings needs to be statistically related to the different seasons.  This could be done with a chi-square test or using a logistic regression model, with the response variable of presence / absence of beachings.  The latter approach would be better, because it would allow you to assess the influence of other variables at once, including inter-annual variability.  Currently, the ms merely reports the %s of summer / winter days with beachings, and a metric of variability (SD for the proportions) is needed  Moreover, these proportions need to be compared statistically, using p values and measures of effect size (like the odds ratio).

* Lags and Multiple Temporal Scales:

While the paper mentions a “zero” lag and provides results at daily and weekly time scales, it is unclear how many lags were tested and how the weekly data were averaged and analyzed.  I would suggest you provide a summary table, showing what analyses were done, listing the lags that were attempted and the different temporal scales that were considered.

* Multi-variate Analyses:

These environmental factors are likely cross-correlated:  wind speed / direction, currents, water temperature.  I would ask the authors to explore these cross-correlations and to provide a supplementary table where these results are summarized.  If there are significant cross-correlations, I would urge the authors to use partial correlations to explore the influence of the drivers, after accounting for other cross-correlated variables. 

Moreover, it would be useful to know whether these environmental drivers differed seasonally and from year-to-year (within seasons).  This would provide the readers with a broader oceanographic background of the study area and the potential drivers. 

Finally, I would also suggest you summarize the weather (wind / current) and water temperature conditions measured during periods of unusually high and unusually low beaching (and stringing) periods.  This would provide a complementary perspective to the previous modeling approach, which would give readers a more in-depth understanding of the drivers of unusual “events”.

We look forward to receiving your revised manuscript.

Kind regards,

David Hyrenbach, Ph.D.

Academic Editor

PLOS ONE

Journal Requirements:

2a) If there are ethical or legal restrictions on sharing a de-identified data set, please explain them in detail (e.g., data contain potentially sensitive information, data are owned by a third-party organization, etc.) and who has imposed them (e.g., an ethics committee). Please also provide contact information for a data access committee, ethics committee, or other institutional body to which data requests may be sent.

2b) If there are no restrictions, please upload the minimal anonymized data set necessary to replicate your study findings as either Supporting Information files or to a stable, public repository and provide us with the relevant URLs, DOIs, or accession numbers. For a list of acceptable repositories, please see http://journals.plos.org/plosone/s/data-availability#loc-recommended-repositories.

3. We note that Figure 1 in your submission contain satellite images which may be copyrighted. All PLOS content is published under the Creative Commons Attribution License (CC BY 4.0), which means that the manuscript, images, and Supporting Information files will be freely available online, and any third party is permitted to access, download, copy, distribute, and use these materials in any way, even commercially, with proper attribution. For these reasons, we cannot publish previously copyrighted maps or satellite images created using proprietary data, such as Google software (Google Maps, Street View, and Earth). For more information, see our copyright guidelines: http://journals.plos.org/plosone/s/licenses-and-copyright.

4. Please include captions for *all* your Supporting Information files at the end of your manuscript, and update any in-text citations to match accordingly. Please see our Supporting Information guidelines for more information: http://journals.plos.org/plosone/s/supporting-information.

Reviewers' comments:

Reviewer's Responses to Questions

**Comments to the Author**

1. Is the manuscript technically sound, and do the data support the conclusions?

Reviewer #1: Partly

Reviewer #2: Yes

2. Has the statistical analysis been performed appropriately and rigorously? 

Reviewer #1: No

Reviewer #2: Yes

3. Have the authors made all data underlying the findings in their manuscript fully available?

Reviewer #1: No

Reviewer #2: Yes

4. Is the manuscript presented in an intelligible fashion and written in standard English?

Reviewer #1: No

Reviewer #2: Yes

5. Review Comments to the Author

Reviewer #1: Dear Author,

You have made a great work into collecting and preparing a several-year dataset of stranded Physalia physalis colonies in three beaches in Australia. However, because they represent just three specific geographical points with their owns costals dynamics (in sence of oceanography and meterology), I recomend you to look forward a journal with emphasis in local studies.

Moreover, you haven't perform the proper statistical analysis to sustain your conclusions. Even you have cited very good papers dealing with the analytical approach required to understand environmental forcing on straned Physalia colonies, you have used just person correlations in just some statements. You must to improve this section (by including a whole Data Analysis section in methods, for instance).

The remaining of the manuscript is based on this (non proved) correlation and forcing and by this make it impossible to approve.

The relationship with the stings again is lacking of the proper statistical approach and must be reformulated.

Reviewer #2: SOME COMMENTS ABOUT THE PAPER:

After reading the article, I write down some recommendations and some thoughts that could help the authors to tweak some comments made throughout the article.

The authors use some apostrophes in grammatical structures where their use at the scientific article level may not be necessary or these structures can be rewritten. It is recommended to review its use with a native speaker, such as in:

…P.physalis's morphology

…P. physalis's course

Line 67:

For validation, these models were compared to massive beaching events that occurred in summer 2010 off the Basque coast (France) and the Mediterranean Basin.

The Basque coast is in Spain, in the autonomous community of the Basque Country. This community ends at the border between Spain and France. In the summer of 2010, the presence of Physalia occurred at several beaches along the coast. But on some important beaches, such as La Concha Beach (in the city of Donostia-San Sebastián), small fishing boats were transformed into cleaning boats and left the beach area to meet the Portuguese man-of-war and collect them before their arrival along the beach.

Line 194:

The few number of sightings in winter as well as P. physalis supposed lifecycle could be explained by a collective death in winter.

For me, a low or no number of winter sightings in the coastal area does not mean that there are a large number of deaths in the open sea. It is to be expected that there are always Portuguese man-of-war of different ages drifting for months in the great oceanic gyres (using the wind as main driver) with a peak of reproduction that could occur at the end of summer-beginning of autumn. On the European coast of the North Atlantic Ocean, it is typical that during the winter (not only in the months of summer) there is also a notable presence of small Physalia (3-5 cm long float, 3-4 months old) that due to the very strong southerly winds of successive storms (favourable to dragging towards the Bay of Biscay) have caused the appearance of Physalia on the coast to be anticipated. Therefore, the highest mortality possibly occurs when these organisms reach the dry beach, where they no longer leave and end up dying. These organisms do not appear to die from the severe winter conditions, at least in the North Atlantic Ocean. These conditions can make it possible for them to reach the coast at a time other than summer. It is for this reason that it is important to monitor the beaches outside of the time that the beaches are patrolled and during the lifetime of the organisms.

Line 348:

[11] similarly suggest that wind is a dominant driver of P. physalis transport, but propose wind to be more relevant offshore and ocean circulation becoming the main driver in nearshore areas.

[11] suggests that the wind is the most relevant mechanism both off and on the coast for this peculiar organism. The very superficial ocean circulation (considering this as the one that exists in the first 5 centimetres of the water column, where Physalia lives) in the great gyres of ocean circulation is greatly influenced by the wind, as shown by very low-weight drift buoys floating on the surface. The data of these buoys shows that the surface ocean circulation is far from following the Ekman theory (that is, generating a surface current at 45 degrees from the wind). It is for this reason that possibly the best solution to explain the drift of Physalia is to use the wind, because also the wind is the generator of local waves and the circulation at the upper centimetres of the water column.

Line 358:

In addition, there was a high frequency of beaching events in spring recorded during weeks that were dominated by south-westerly winds, as can be observed during September and May in Fig 5. This result is surprising since beachings would not be expected when wind is coming from land, if wind were the only driving variable.

To study the arrival of these organisms, it would be necessary to analyse not only the winds of the days prior to arrival, but also the evolution of winds throughout the life of these organisms, which could be from a few months to a year (more or less), depending on the size of the organism. Prevailing southwesterly winds could probably bring many Physalia located in the open sea below Australia. And winds from the northeast, east or southeast (in the days prior to arrival), even if they were of short duration, could cause these organisms to end up in the study beaches. Therefore, it is highly recommended to analyse the annual evolution of the wind in a very large area (several degrees in longitude and latitude) around the study area. Surely these organisms have been able to travel more than 10,000 kilometres on their journey to reach the beach.

Line 371:

Observations of ocean currents closer to the surface and of higher resolution (e.g. coastal High-Frequency RADAR) may be necessary to expose any dependence of beaching events on these variables.

The fundamental problem with using high-frequency radar observations to explain caravel drifts is that they provide information on currents at 1-3 meters above the surface. This information is quite different from that existing in the same ocean-atmosphere interface, that is, in the first centimeters of the water column. So to speak, the Portuguese caravel is a very light balloon (a caravel of 10 centimeters of float can weigh around 25 grams) that has tentacles that act as an anchor so that it does not fly. So it seems unlikely that trying to explain their drift with currents below 5-10 centimeters from the sea surface will not do much.

6. PLOS authors have the option to publish the peer review history of their article (what does this mean?). If published, this will include your full peer review and any attached files.

Reviewer #1: No

Reviewer #2: No

---

## [Author Response · Author response to Decision Letter 0]

24 Sep 2021

21/09/2021

Dear editor,

We thank you for the opportunity to revise manuscript ID PONE-D-21-06875 entitled “Driving the

blue fleet: Temporal variability and drivers behind bluebottle Physalia physalis beachings off

Sydney, Australia” which was submitted for consideration for publication in PLOS ONE.

We thank the reviewers for their thoughtful suggestions to our study. We have addressed our

responses to the reviewers' comments below and identified where changes have been made in the

revised manuscript using track changes and here with green text. We hope that the study is now

suitable for publication. It would be an honour to be published in PLOS ONE.

Yours sincerely,

Natacha Bourg and co-authors

REVIEWER 1:

Comments on:

“Driving the blue fleet: Temporal variability and drivers behind bluebottle Physalia

physalis beachings off Sydney, Australia.”

Abstract

Reviewer comment: The usage of “Jellyfish-like colonial organism” is not required since the formal

definition of Jellyfish encompasses the order Siphonophorae.

Author response: We have changed to “are colonial siphonophores that live at the surface of the

ocean” (Line 1 of Abstract).

The are other datasets of Physalia strandings covering comparable temporal coverage, so the use of

the words “unprecedented dataset”.

We did not come across such a daily presence / absence dataset for 4 years (please provide the

reference) but “unprecedented" was removed.

Classical presence/absence related with the higher sampling effort in summer?, in abstract must

include the avoidance to this factor.

The two sites focused on in this study (Maroubra and Coogee) have the same sampling rate (daily)

for presence / absence (lifeguard observations) in summer and throughout the year.

Introduction

Line 7: Reference #2 must be updated since the provided reference link to the pre-print version, but

now the work from Munro et al., can be found here (https://doi.org/10.1038/s41598-019-51842-1).

Thank you, the reference has been updated line 7.

Line 8: A reference must be given to this statement.

References have been added line 9 regarding the predatory behaviours and diet of P. physalis.

Specifically, Purcell (1984) and Munro et al. (2019) have been added.

Line 12: The cited reference (Prieto et al., 2015) doesn't account for any Fatal encounter with P.

physalis. The proper citation must be given.

We have added proper citation: Burnett et al (1989) line 13.

Line 20: You must provided already available (and listed in your reference list) information dealing

with spatio-temporal distribution and drivers of massive stranding, for this species in other places.

Line 20-21: There isn’t such a gap of information. Proper references must be given (see work from

Pontin et al., Canepa, et al, etc). You can refine the knowledge to some specific location and/or

analytical process; but the meaning of a gap isn’t what we have today.

We have completely re-written and organised the introduction, including a paragraph on the spatiotemporal

context and previous studies. We had missed the study from Canepa et al. (2020) and are

grateful for the mention of this great study. The new paragraph read ( Lines 40-71):

“Previous studies have linked P. physalis beach stranding to environmental conditions and model

their arrival to the coast, but have usually focused on unusual events (e.g. swarms). Ferrer and

Pastor (2017); Prieto et al. (2015) studied massive beaching events that occurred in summer 2010

off the Basque coast (Spain) and the Mediterranean Basin using lagrangian particle tracking. Ferrer

and Pastor (2017) proposed that offshore origin of these beached P. physalis was strongly dependent

on the wind parametrisation used in the lagrangian tracking model. They therefore concluded these

beaches P. physalis, were likely to have originated from the northern part of the North Atlantic

Subtropical Gyre, thousands of kilometers away. They proposed wind as a dominant driver of P.

physalis transport both off and on the coast. Prieto et al. (2015) suggested that the massive arrival of

P. physalis to the coast had been strongly influenced by zonal winds. Since then, massive beachings

of P. physalis off the coast of Ireland in autumn 2016 (August and October) prompted further

research (Headlam, 2020) to identify source populations of P. physalis. Results suggested that the

population of P. physalis may have originated from the North Atlantic Current, supporting the

findings of Ferrer and Pastor (2017). Using sting reports from five summers across eight locations

in New Zealand, Pontin et al. (2009) developed a neural network-based model to simulate the

arrival of P. physalis towards the shore while assessing the contribution of large-scale winds and

waves. They show that wave direction far from the shore can transport swarms to the studied

region, while wind direction and speed one day before sting reports, in cells close to the shore

strongly influence the strandings of P. physalis. A more recent and similar study by Pontin et al.

(2011) extended the analysis to different regions all around New-Zealand. They validate results

found by Pontin et al. (2009) that both wind and wave influence the occurrence of P. Physalis

strandings, while highlighting that different oceanographic regimes driving the beachings occur in

different study areas. Unlike previous studies, a recent survey from Canepa et al. (2020) recorded

nearly continuous strandings of P. Physalisin Chile for three years, with the highest densities in the

winter seasons two years in a row. These massive events coincided with an El Ni ~no Southern

Oscillation (ENSO) perturbation, with warmer ocean temperature conditions, and positive zonal

wind anomalies (westerlies) transporting P. physalis to the coast, further highlighting wind as the

major driver of P. physalis movements and variability

”

Line 28-30: You use a moon’s phase effect over monthly aggregation of a box jellyfish which is

totally different from a “pleustonic drifter” as Physalia physalis is. Beside this, you don’t provide

any reference and/or data that supports this statement. Please modify.

Thank you for this comment. However, the moon phase has been suggested to be relevant for P.

physalis in Hawaii. Even so, we have removed all analyses and mentions to the moon cycle in the

manuscript since we observed no association.

Line 30-31: You suggest a physiologic response of P. physalis to the moon cycle, based in one

statement which hasn’t any support. Remove this phrase.

The sentence has been removed.

Line 69: The Basque coast as cited in the proposed reference (“individuals of this species arrived at

the Basque coast (southeastern Bay of Biscay)”) refers to Spain and not France.

Our apologies, this has been rectified, line 42, and now reads:

“Prieto et al., 2015 and Ferrer and Pastor, 2017 studied massive beaching events that occurred in

summer 2010 off the Basque coast (Spain) and the Mediterranean Basin using lagrangian particle

tracking.”

Line 88: The reference here (Pontin et al.) must have the number 14 and not the number 21. Beside

this, there is another report from the same author which is much conclusive and required to be

incorporated into the document (refer to this: https://doi.org/10.1016/j.ecolmodel.2011.03.002).

Thank you for this information, we have updated these references as suggested see line 61.

Methods

Fig. 1: The satellite image isn’t clear enough where comes from. A dashed square highlighting the

area (or something similar) will be required.

Thank you, the image has been modified to illustrate this (see image below).

Line 137-138: Provide the statistical summary of the no-correlation between number of sting and

beach-goers’ statement.

Nothing is said about how author’s will analyze the effects of environmental variables over the

stranded data. A full section of Data Analysis is highly recommended, since this step is crucial in

the development of the objectives.

Thank you for highlighting this, there is now a full section with the heading Statistical Analyses in

the revised manuscript (lines 162-180) where a new statistical analysis is presented, and the

methods are described in detail.

Results

Line 193 and general: When highlighting comparisons about the number and/or percentage the

statistical result must be given.

The Results section has almost been entirely re-written including new statistical analyses. Please see

details below.

Line 196: There is no enough evidence along the section to infer a collective death in winter, since

some colonies can be washed offshore by the change in wind conditions. Also the life span of

colonies is highly unknown.

This statement has been modified, and now reads (lines 202-206):

"Interestingly, there are still instances of winter beaching for Coogee and Maroubra, up to 10% of

annual sightings in Coogee. This suggests that despite the seasonal cycle to their strandings, P.

physalis survive wintertime and cold temperatures and can still be advected to the coast.”

Drivers of beaching events

Line 202-203: No formal statistical approach is given to give support to that statement.

Line 204: Just for the wave height condition the usage of the Pearson’s correlation coefficient is

showed but considering the literature that authors have read, they should know that a direct

correlation coefficient isn’t enough to explain complex spatio-temporal process.

Line 210-225: The association between environmental variables and stranded jellyfish are analyzed

individually and mostly based on visual inspection of the environmental and biological process,

without the usage of proper statistical approaches.

The revised manuscript now includes a whole new set of statistical analysis. We have added lagged

correlations, and results from Generalized Estimating Equations (GEE) models for the two allyearlong

beaching sites, and different timescales (daily and weekly), that support our previous

findings. Please see Data & Methods (lines 163-180) and Results (lines 220 and lines 252).

Seasonality of environmental variables

Line 230: There is no prove to sustain that statement

In the new GEE analysis, we investigate a possible relationship between the different environmental

variables and beaching events, supporting the statistical relationship between zonal (I.e., crossshore)

wind and beaching events, see Section Results lines 270-278 and Table 2.

Line 308: Put a comma after “Interestingly”

Done, line 315.

Discussion

Lines 319-321: You cannot sustain this statement “Our results demonstrate a strong relationship

between this spatio-temporal variability and wind direction at the daily timescale”, since you

haven't provided an analytical framework where statistical-based conclusions have arise. This

requires much more than image (visual) inspection.

We agree with the reviewer, and hope that the new statistical analysis mentioned above will

convince them.

Lines 323-325: You have not inspect the proportion of left/right-handed stranded Physalia physalis

colonies in your study; so you cannot conclude a direct effect from the wind, avoiding local surface

currents in complex coastal areas as you have signaled.

The reviewer is right, this was only a supposition which needs to be tested when the relevant

observations are available. This is what we meant by “It is possible that” at the beginning of the

sentence.

Conclusion

In general conclusions must be re-written after the proper analytical method have been given and

executed.

The whole manuscript has been re-written including a better organized introduction, new analytical

methods and results, which provide additional support to our conclusion.

REVIEWER 2:

SOME COMMENTS ABOUT THE PAPER:

After reading the article, I write down some recommendations and some thoughts that could help

the authors to tweak some comments made throughout the article.

We are thankful for the constructive comments which helped to improve the manuscript. Please find

details below.

The authors use some apostrophes in grammatical structures where their use at the scientific article

level may not be necessary or these structures can be rewritten. It is recommended to review its use

with a native speaker, such as in:

…P.physalis's morphology

…P. physalis's course

Updated to P. physalis, thanks.

Line 67:

“For validation, these models were compared to massive beaching events that occurred in summer

2010 off the Basque coast (France) and the Mediterranean Basin.”

The Basque coast is in Spain, in the autonomous community of the Basque Country. This

community ends at the border between Spain and France. In the summer of 2010, the presence of

Physalia occurred at several beaches along the coast. But on some important beaches, such as La

Concha Beach (in the city of Donostia-San Sebastián), small fishing boats were transformed into

cleaning boats and left the beach area to meet the Portuguese man-of-war and collect them before

their arrival along the beach.

Our apologies, this has been rectified (line 43). Thanks for sharing the background for this event, it

is nice to see that this was a shared effort between different communities.

Line 194:

“The few number of sightings in winter as well as P. physalis supposed lifecycle could be explained

by a collective death in winter.”

For me, a low or no number of winter sightings in the coastal area does not mean that there are a

large number of deaths in the open sea. It is to be expected that there are always Portuguese man-ofwar

of different ages drifting for months in the great oceanic gyres (using the wind as main driver)

with a peak of reproduction that could occur at the end of summer-beginning of autumn. On the

European coast of the North Atlantic Ocean, it is typical that during the winter (not only in the

months of summer) there is also a notable presence of small Physalia (3-5 cm long float, 3-4

months old) that due to the very strong southerly winds of successive storms (favourable to

dragging towards the Bay of Biscay) have caused the appearance of Physalia on the coast to be

anticipated. Therefore, the highest mortality possibly occurs when these organisms reach the dry

beach, where they no longer leave and end up dying. These organisms do not appear to die from the

severe winter conditions, at least in the North Atlantic Ocean. These conditions can make it possible

for them to reach the coast at a time other than summer. It is for this reason that it is important to

monitor the beaches outside of the time that the beaches are patrolled and during the lifetime of the

organisms.

We have removed this statement and thank the reviewer for the additional examples. We have also

observed P. physalis in winter, but much less than in summer, even during favourable wind

conditions. We are planning a survey of P. physalis‘ size to investigate the link between size and

seasonality in the area, all year-round. We would be interested in following up this comment

directly if possible.

Line 348:

“[11] similarly suggest that wind is a dominant driver of P. physalis transport, but propose wind to

be more relevant offshore and ocean circulation becoming the main driver in

nearshore areas.”

[11] suggests that the wind is the most relevant mechanism both off and on the coast for this

peculiar organism. The very superficial ocean circulation (considering this as the one that exists in

the first 5 centimetres of the water column, where Physalia lives) in the great gyres of ocean

circulation is greatly influenced by the wind, as shown by very low-weight drift buoys floating on

the surface. The data of these buoys shows that the surface ocean circulation is far from following

the Ekman theory (that is, generating a surface current at 45 degrees from the wind). It is for this

reason that possibly the best solution to explain the drift of Physalia is to use the wind, because also

the wind is the generator of local waves and the circulation at the upper centimetres of the water

column.

We have removed the sentence and agree with the reviewer, as mentioned in the Discussion: “It is

important to highlight that the wind is a major contributor to the ageostrophic component of the

surface current (influencing circulation and generating local waves) and stokes drift and windinduced

currents are known to be highly relevant in regard to the transport of passive tracers,

especially in the first centimeters of the ocean. Then, this relationship between beaching events and

cross-shore wind can be explained by the wind drag on P. physalis outside of water, but also by the

action of wind-induced current on its transport”. (Lines 357-363)

Line 358:

“In addition, there was a high frequency of beaching events in spring recorded during weeks that

were dominated by south-westerly winds, as can be observed during September and May in Fig 5.

This result is surprising since beachings would not be expected when wind is coming from land, if

wind were the only driving variable.”

To study the arrival of these organisms, it would be necessary to analyse not only the winds of the

days prior to arrival, but also the evolution of winds throughout the life of these organisms, which

could be from a few months to a year (more or less), depending on the size of the organism.

Prevailing southwesterly winds could probably bring many Physalia located in the open sea below

Australia. And winds from the northeast, east or southeast (in the days prior to arrival), even if they

were of short duration, could cause these organisms to end up in the study beaches. Therefore, it is

highly recommended to analyse the annual evolution of the wind in a very large area (several

degrees in longitude and latitude) around the study area. Surely these organisms have been able to

travel more than 10,000 kilometres on their journey to reach the beach.

We agree that the environmental conditions during P. physalis journey will determinate its

trajectory, including how likely it will be close to the coast. However, to reach and strand on the

beach, we show that the last 24 hours are key. While the long-term large-scale drivers are definitely

of interest, we would need to know where P. physalis comes from and how long it has been floating

in the ocean, which is still largely unknown. This will be the topic of further investigation in the

future.

Line 371:

“Observations of ocean currents closer to the surface and of higher resolution (e.g. coastal High-

Frequency RADAR) may be necessary to expose any dependence of beaching events on these

variables.”

The fundamental problem with using high-frequency radar observations to explain caravel drifts is

that they provide information on currents at 1-3 meters above the surface. This information is quite

different from that existing in the same ocean-atmosphere interface, that is, in the first centimeters

of the water column. So to speak, the Portuguese caravel is a very light balloon (a caravel of 10

centimeters of float can weigh around 25 grams) that has tentacles that act as an anchor so that it

does not fly. So it seems unlikely that trying to explain their drift with currents below 5-10

centimeters from the sea surface will not do much.

The reviewer is right, the significant vertical shear at the surface makes HF radars not optimal. We

have removed this statement.

EDITOR :

Thank you for submitting your manuscript to PLOS ONE. After careful consideration, we feel that

it has merit but does not fully meet PLOS ONE’s publication criteria as it currently stands.

Therefore, we invite you to submit a revised version of the manuscript

that addresses the points raised during the review process.

Before this ms can be published, substantial changes need to be implemented to augment the

analyses and strengthen the main lessons of the paper.

We thank the editor for their time and effort to provide additional comments which helped us

improving the manuscript. We hope that the editor will agree that the revised manuscript provides

the statistical evidence to support our conclusion. We also made substantial changes to the writing

and organization of the manuscript.

The introduction needs to be streamlined. For instance, the second paragraph (lines 17 to 46) is very

lengthy and could be split into smaller focused sections. The next two paragraphs (lines 47 to 94)

could also be streamlined, and some of the material could be moved to the discussion, where it

would be placed in context of the project’s findings.

The Introduction as well as most other sections of the manuscript have been re-written, taking into

consideration all comments.

As stated by one reviewer, the authors have not performed the proper statistical analysis to sustain

their conclusions. Even though several papers dealing with the analytical approach required to

understand environmental forcing are cited, the authors have only used person correlations in their

analysis. The authors have to improve this section (by including a whole Data Analysis section in

the methods) and undertake a more comprehensive analysis of the data at hand.to determine the

influence of these factors (and potentially their interactions) on the beachings (and the summer

stings). The current piece-meal approach, where a single variable is considered at a time need to be

augmented and strengthened.

New analyses are proposed in the revised manuscript. We chose to use a Generalised Estimating

Equations (GEE) model, taking into account multiple explanatory variables and using the daily

beaching dataset as the response. We used an autoregressive AR(1) structure for the GEE model to

consider strong auto-correlations in the daily time-series. The data at the two sites which have

observations all year-round were considered, not the site which is only patrolled in summer. See

Data & Methods (lines 163-180) and Results (lines 220 and lines 252).

To facilitate the understanding of the patterns, I would also suggest focusing on the two sites with

year-long data and removing the third site (rocky shore with only summer-time data). Limiting the

analysis to the two sites with year-long data (Clovelly and Maroubra) provides a more

comprehensive and comparable perspective. The ms already explains that this site is inherently

different: “Note that Coogee has a small rocky outcrop (known as the Wedding Cake Island) 740 m

from the beach, which limits wave action on the beach. Clovelly beach is more South-oriented and

is at the end of a narrow bay, hence more protected than the two other beaches (Fig 1).”

There is a misunderstanding between Coogee and Clovelly here. Coogee and Maroubra cover all

seasons and were the focus of statistical analysis. Still, we mostly use Maroubra as it seems to be

the most “neutral” (fewer missing data, largest beach with a quite neutral orientation, no island

located in front of the beach). Clovelly data, which is patrolled only in summer, is only shown when

focusing on summer.

Additionally, the Physalia physalis datasets need to be analyzed in a more quantitative fashion. In

particular, I would suggest the following analyses:

* Number of beachings:

Compare the number of beaching observations versus the number of survey days from a beach to

beach. There are 38 and 132 beaching reports for Clovelly and Maroubra respectively, even though

the two beaches were surveyed on 94% and 93% of the days. Is this difference significant? Is there

an overall higher beaching rate in Maroubra? Despite the data gaps, I would suggest you perform a

cross-correlation to quantify how well the beachings data at the two beaches cross-correlate with

each other.

We have added the information in the manuscript. It now read lines 192-195: “Simultaneous

beaching in Maroubra and Coogee occurs only 10-20% of the beaching days, and the correlation

between the timeseries of beaching presence at the daily timescale is r = 0.1, increasing to r = 0.3

when considering the weekly number of beachings.”

* Number of stings:

I would suggest focusing this analysis on the same two beaches used in the beachings analysis, and

discarding the data from Coogee. Despite the data gaps, I would suggest you perform a crosscorrelation

to quantify how well the beachings data at the two beaches cross-correlate with each

other.

You state that “More than 10 stings have been reported 6, 9 and 10% of all patrolled days for

Clovelly, Coogee and Maroubra”.

Why did you not consider days where less than 10 stings have been recorded? You could use values

above and below this threshold as two separate categories (low and high), or you could take the

log10-transform of the data.

How was this threshold number selected? Seems like anomalous events should be determined on a

beach-basis, not using the same threshold across all beaches. I would suggest you provide a data

summary of the number of stings reported per day, and then attempt to model these distributions to

figure out “outlier days” for each beach.

* Number of beachings VS Number of stings:

It would be very useful to investigate whether these two datasets are correlated. Using the summerperiod

only, when stings are reported, can you perform a correlation for each beach, to see if there

are more stings on days with more beachings. This would be a very informative analysis.

We have now included more information on the comparison between beaching and stings (lines

118-122): "Comparing the beaching and sting datasets for matching days and locations (although

different authors), only 7.9 %, 15.8%, and 32.3% of the stings corresponded to a day when beaching

was also at Clovelly, Coogee, and Maroubra, respectively. The daily match between these two

datasets needs more investigation (see Discussion) and longer time-series.”

The threshold of 10 stings have been chosen to remove outliers, since we suspect that the lifeguards

do not report the presence of P. physalis when they only see one specimen. We have performed a

sensitivity study to this threshold, but the correspondence does not change much.

For your interest, the two figures below provide additional details, including the beach attendance,

and the ratio between stings/beach attendance (size of the symbols). The association between stings

and beach attendance might exist in Coogee, but is not clear in Maroubra. Furthermore, even days

with high numbers of stings do not necessarily correspond to a beaching observation (see the

colours).

We discuss the potential reasons for the discrepancy between stings and beaching in the manuscript,

but this will need further investigation (lines 373-375):

"Differences between the two datasets could be explained by the difference in the timing of the

reports but also by the nature of the reports (stings happen in the water, while beachings are

reported only when P. physalis are stranded on the shore).”

Figure caption: Scatter plot showing beach

attendance against number of stings, with the colours displaying wether the beaching was reported

by lifesavers or not for the same day. The size of the markers show the percentage of stings

compared to the beach attendance.

* Wind Data:

Can you please define the wind sectors and provide some summaries of wind speed / direction for

the different seasons? The ms currently states “predominant winds in this area are north-easterly,

westerly and southerly, as shown on the windrose in Fig 1.”

The analyses of beachings per wind direction also need to involve statistical tests, using either chisquare

tests or logistic regressions. Reporting mere proportions is not enough. You need to provide

a sense of the variability (SD for the proportions) and the associated p values.

Instead of a logistic regression, we included a GEE model (see comment above), which takes into

account the variability of the wind. We also refer to Wood et al (2016) which details the monthly

mean and variance of the same wind dataset (see line 144).

* Ocean Currents:

Can you please report how well the near-surface and the integrated currents correlate with each

other? And report how well they match the wind speeds? Currently, the ms states: “Here, we used

daily averages at the shallowest bins (11 m and 12 m, respectively) and the depth integrated flow”.

Unfortunately, the shallowest surface current estimates we have in the region are at 11m and 12m,

which we expect to be less wind-driven than the top few centimeters of the ocean.

For your information, the figure below shows the correlation matrix the zonal (u) and meridional (v)

components of the 11m-depth and depth-averaged currents, and the wind velocity. It shows that

depth-integrated ocean currents are strongly correlated to the wind, but less so for the 11m depth

current.

Figure caption :

Correlation matrix of

different variables: wind

(u_wind, v_wind), depthintegrated

current from the

mooring depth-integrated

(u_cur_rot_int,

v_cur_rot_int), and at 11m

(u_cur_rot_11m,

v_cur_rot_11m).

* Seasonality:

The proportion of

beachings needs to be

statistically related to the

different seasons. This could be done with a chi-square test or using a logistic regression model,

with the response variable of presence / absence of beachings. The latter approach would be better,

because it would allow you to assess the influence of other variables at once, including inter-annual

variability. Currently, the ms merely reports the %s of summer / winter days with beachings, and a

metric of variability (SD for the proportions) is needed Moreover, these proportions need to be

compared statistically, using p values and measures of effect size (like the odds ratio).

The variability of beaching is shown in Figure 6, as a shading around the weekly means.

The inter-annual variability of the presence/absence of beaching is included in the GEE model as

the response variable, and the p-values reported take into account the auto-correlation of the timeseries.

See comments above.

* Lags and Multiple Temporal Scales:

While the paper mentions a “zero” lag and provides results at daily and weekly time scales, it is

unclear how many lags were tested and how the weekly data were averaged and analyzed. I would

suggest you provide a summary table, showing what analyses were done, listing the lags that were

attempted and the different temporal scales that were considered.

Thank you for the remark, we have added Figure 3 in the manuscript to show correlations at

different lags, supporting the choice of investigating the wind with a 24h lag.

Figure caption : Pearson correlation coefficient between beaching events at Maroubra anddifferent

environmental variables, for different lags. A negative lag means consideringvariables a day before

the beaching day

* Multi-variate Analyses:

These environmental factors are likely cross-correlated: wind speed / direction, currents, water

temperature. I would ask the authors to explore these cross-correlations and to provide a

supplementary table where these results are summarized. If there are significant cross-correlations,

I would urge the authors to use partial correlations to explore the influence of the drivers, after

accounting for other cross-correlated variables.

New analyses answer to this as GEE is multivariate.

Moreover, it would be useful to know whether these environmental drivers differed seasonally and

from year-to-year (within seasons). This would provide the readers with a broader oceanographic

background of the study area and the potential drivers.

We have not considered inter-annual variability in this study since four years is a relatively short

time-scale. We have however shown how ocean temperature, wind speed, and cross-shore winds

vary with seasons in Figure 6. For broader background, we refer to Wood et al. (2016), in particular

their Figure 5 which shows how winds and currents in the region vary month to month.

Finally, I would also suggest you summarize the weather (wind / current) and water temperature

conditions measured during periods of unusually high and unusually low beaching (and stringing)

periods. This would provide a complementary perspective to the previous modeling approach,

which would give readers a more in-depth understanding of the drivers of unusual “events”.

We agree that understanding unusual events is of great interest. This is discussed in the discussion

in a whole paragraph, lines 377-394.

Unfortunately, we do not have the right observational dataset to investigate properly the influence

of these parameters, hence cannot provide more than an educated guess for these events.

---

## [Editor Report · Decision Letter 1]

18 Nov 2021

PONE-D-21-06875R1Driving the blue fleet: Temporal variability and drivers behind bluebottle Physalia physalis beachings off Sydney, Australia.PLOS ONE

Dear Dr. Bourg,

Thank you for submitting your manuscript to PLOS ONE. After careful consideration, we feel that it has merit but does not fully meet PLOS ONE’s publication criteria as it currently stands. Therefore, we invite you to submit a revised version of the manuscript that addresses the points raised during the review process.

The authors have made substantial changes to the manuscript, but there are many pending issues that need to ne addressed, before this manuscript can be published.    

The main issues that need to be addresses are:

1. The paper describes analyses, but does not provide the necessary details to interpret the results

Line 111:“We explored whether sting numbers were dependent on numbers of beachgoers, but found no clear correlation between the two” Can you please explain how this was done and what were the results?

Line 121:“The daily match between these two datasets needs more investigation”. These statements also need more detailed reporting of the analyses. It is not possible to assess what was done and what is the correspondence between “stings” and “beachings”.

2. There are several untested assumptions for the statistical tests. 

Line 169:

NOTE:

The binary response data are not normal - but binomial.

The rip currents are rank data, not numerical.

3. Some of the discussion of the results compare proportions or discuss similarities, without actually performing the statistical analyses.   For instance:  Line 192:Did you attempt crosscorrelations between the two sites? This would be a much better way to assess covariability.

Table 3 and Table 4:  Can you please compare the observed proportions versus the expected proportions?  How frequently are these wind conditions observed, will influence whether these results are significant or not.

Line 198:  “Indeed, between 2016 and 2020, 50% and 46% of strandings occurred during the three months of summer in Maroubra and Coogee respectively. In Maroubra, spring is (after summer) the second season with most beaching events (30% of beachings), whereas in Coogee, beaching events are more numerous in autumn (25%) than spring” 

Can you please perform tests to determine if seasons matter: statistically speaking?  Merely mentioning the proportions of events is not enough to determine whether these proportions are significantly different from what we would expect.  Did you define the seasons equally, so each one accounts for 25% pf the time?  This would et the expected proportions.  But we need a way to assess if these proportions are significantly different from the observed proportions.

4. The writing needs to be organized:  much material needs to be moved:

From Methods to Introduction (see notes in the pdf)From Results to Discussion (see notes in pdf)

5. The writing needs to be improved substantially to streamline the text, clarify the writing and fix some grammar and typos (like the persistent use of “data” in singular).

6. Finally, Page 3 – Figure Caption 1:  Can you please credit the images.

You used images from “Satellite image of the different beaches (From The Gateway to Astronaut Photography of Earth”.  Is this a free creative commons product? 

Can you provide a reference?    I found the site, but there is no information on the use of these images:  https://eol.jsc.nasa.gov/SearchPhotos/

We look forward to receiving your revised manuscript.

Kind regards,

David Hyrenbach, Ph.D.

Academic Editor

PLOS ONE
---

## [Author Response · Author response to Decision Letter 1]

28 Feb 2022

Response to reviewers file is attached to the revision.

---

## [Editor Report · Decision Letter 2]

7 Mar 2022

Driving the blue fleet: Temporal variability and drivers behind bluebottle Physalia physalis beachings off Sydney, Australia.

PONE-D-21-06875R2

Dear Dr. Bourg,

We’re pleased to inform you that your manuscript has been judged scientifically suitable for publication and will be formally accepted for publication once it meets all outstanding technical requirements.  Thank you for addressing all the requested changes and revisions.

Kind regards,

David Hyrenbach, Ph.D.

Academic Editor

PLOS ONE
---

## [Editor Report · Acceptance letter]

9 Mar 2022

PONE-D-21-06875R2 

Driving the blue fleet: Temporal variability and drivers behind bluebottle (*Physalia physalis*) beachings off Sydney, Australia 

Dear Dr. Bourg:

I'm pleased to inform you that your manuscript has been deemed suitable for publication in PLOS ONE. Congratulations! Your manuscript is now with our production department. 

Kind regards, 

on behalf of

Dr. David Hyrenbach 

Academic Editor

PLOS ONE